Prediction of clinical deterioration within one year in chronic obstructive pulmonary disease using the systemic coagulation-inflammation index: a retrospective study employing multiple machine learning method

Hou Ling 1
Min Ming 2
Hou Rui 3
Tan Wei 2
Zhang Minghua 1115295145@qq.com 2
Liu Qianfei l879612253@163.com 2
1 Department of Central Hospital of Tujia and Miao Autonomous Prefecture, Hubei University of Medicine , Hubei , China
2 Department of Pulmonary and Critical Care Medicine, Central Hospital of Tujia and Miao Autonomous Prefecture , Enshi , China
3 Hubei Enshi College , Enshi , China
Upadhyay Rohit
Electronic publication date: 2025 Feb 25
Publication date: 2025
Volume: 13
Electronic Location ID: e18989
Received 2024 Oct 3; Accepted 2025 Jan 22
Copyright: ©2025 Hou et al.
Copyright year: 2025
Copyright holder: Hou et al.
License: This is an open access article distributed under the terms of the Creative Commons Attribution License, which permits unrestricted use, distribution, reproduction and adaptation in any medium and for any purpose provided that it is properly attributed. For attribution, the original author(s), title, publication source (PeerJ) and either DOI or URL of the article must be cited.
License URL: https://creativecommons.org/licenses/by/4.0/

Keywords: Clinical deterioration, Chronic obstructive pulmonary disease, Systemic coagulation-inflammation index, Machine learning, Predictor

Funding: The authors received no funding for this work.

==============================
Background

Inflammatory response and the coagulation system are pivotal in the pathogenesis of clinical deterioration in chronic obstructive pulmonary disease (COPD), prompting us to hypothesize that the systemic coagulation-inflammation (SCI) index is associated with clinical deterioration in COPD.

Methods

A cohort of 957 COPD patients (mean age: 68.4 ± 7.8 years; 74.4% male) from January 2018 to December 2021 was analyzed. Six machine learning models (XGBoost, logistic regression, Random Forest, elastic net (ENT), support vector machine (SVM), and K-nearest neighbors (KNN)) were evaluated using accuracy, precision, recall, F1-score, and the area under the receiver operating characteristic curve (AUC-ROC).

Results

Our study encompassed 957 patients, out of which 171 were classified in the clinical deterioration of COPD (cd-COPD) cohort. Significant disparities in age, comorbidities like respiratory failure, C-reactive protein, lymphocyte count, red blood cell distribution width (RDW), SCI, procalcitonin (PCT), and D-dimer were depicted between the cd-COPD and non-cd-COPD groups. Concerning machine learning and model comparison, the SVM model showcased consistent performance and strong generalization capabilities on both the training and testing sets compared to the other five machine learning (ML) models. The SCI index, as the most influential predictor, demonstrated a median of 93.08 in cd-COPD compared to 81.67 in non-cd-COPD patients.

Conclusion

The SCI is markedly elevated in cd-COPD patients compared to COPD patients, and SVM demonstrates reliable performance in cd-COPD prediction.

Background

Chronic obstructive pulmonary disease (COPD) is a progressive respiratory condition characterized by irreversible airflow limitation (GBD 2019 Chronic Respiratory Diseases Collaborators, 2023). The escalating global incidence and associated mortality rates of COPD present a significant challenge to the healthcare industry worldwide (Raherison & Girodet, 2009). In China, 8.6% of adults are afflicted with COPD, ranking it as the third most prevalent chronic disease following hypertension and diabetes (Wang et al., 2018). Projections indicate that by 2030, COPD will rank seventh on the global burden of disease index (Labaki & Rosenberg, 2020). The rising prevalence of COPD has led to substantial economic implications and public health concerns. clinical deterioration of COPD (cd-COPD) signifies a critical phase in the disease course marked by severe cough, dyspnea, and the potential for respiratory failure and mortality, contributing to heightened death rates (Ritchie & Wedzicha, 2020). Early prediction of cd-COPD risk, prompt identification of high-risk populations, and accurate prognosis evaluation are imperative for the clinical care of COPD patients.

The clinical deterioration of COPD is linked to escalated systemic and airway inflammatory responses, with increased inflammation exacerbating clinical symptoms and impairing lung function in individuals (Welte, 2014). Apart from accentuated systemic inflammation, cd-COPD can trigger activation of the coagulation system (Mkorombindo & Dransfield, 2021). Emerging evidence indicates that both inflammatory responses and the coagulation system play pivotal roles in the onset and progression of cd-COPD, influencing host immune modulation (Bazzan et al., 2023). For example, data from Liu et al. (2011) in COPD patients suggest that elevated C-reactive protein and fibrinogen levels are closely associated with increased disease severity and risk of clinical deterioration, highlighting the interplay between inflammation and coagulation. Building on this foundation, recent investigations into the systemic coagulation-inflammation index (SCI), an innovative hematological marker, have shed light on its potential utility in reflecting the complex dynamics between coagulation disorders and inflammation. Current research has demonstrated an association between SCI and complications such as bleeding in patients with acute coronary syndrome and those undergoing surgery for acute type A aortic dissection. Despite these findings, its clinical application remains limited (Liu et al., 2022; Zengin & Severgün, 2023). This study represents the first attempt to utilize SCI for evaluating the risk of clinical deterioration in COPD patients within a one-year period. Specifically, the systemic coagulation-inflammation index, an innovative hematological marker, effectively mirrors the dynamics of coagulation disorders and inflammation.

Thus, we hypothesize that there is a direct correlation between elevated SCI levels and the occurrence of cd-COPD, offering a promising avenue for risk stratification and targeted intervention in this vulnerable patient population.

Methods

Study population

The data was collected from January 2018 to December 2021 on COPD diagnosed patients admitted to our hospital, identified by International Classification of Diseases, 10th Revision (ICD-10) codes J440, J441, J449. Clinical deterioration was defined as at least one acute exacerbation event requiring hospitalization within one year after discharge. Patients were ultimately categorized into the cd-COPD group and non-cd-COPD group. Exclusion criteria included immune, or psychiatric illnesses; presence of malignancies, hematologic disorders, or hepatic/renal dysfunction; and loss to follow-up (Fig. 1). This retrospective study adhered to the Helsinki Declaration, received approval from the Ethics Committee of Enshi Autonomous Prefecture Central Hospital (Ethical Approval Number:2024-053-01).All patient data utilized in this article has been thoroughly anonymized before access and analysis, ensuring the elimination of potential risks to individual patients or their personal privacy. Consequently, the requirement for patient informed consent has been waived.

Figure 1 Study flowchart.

Data collection

Clinical data, encompassing demographic details (gender, age), comorbidities (e.g., hypertension, diabetes, coronary heart disease), and various laboratory parameters (complete blood count, random venous glucose, lipid profile), were extracted from the electronic medical records. Treatment details during hospitalization, such as mechanical ventilation and length of stay for initial COPD diagnosis, were also included. The systemic coagulation-inflammation index was calculated using the formula: (platelet count * fibrinogen)/white blood cell count upon admission.

Statistical analysis

Statistical analysis was conducted using R 4.3.2 (R Core Team, 2023) and Python 3.9. Student’s t-test was utilized for normally distributed continuous variables, while the chi-square test or Fisher’s exact test was employed for categorical variables. Skewed data were described using the interquartile range (IQR) and assessed using the Mann–Whitney U test.

Model implementation

Six machine learning (ML) algorithms were utilized, namely, XGBoost, logistic regression, Random Forest, elastic net (ENT), support vector machine (SVM), and k-nearest neighbors (KNN). These models are widely used in clinical research for outcome prediction, as supported by previous studies (Pantanowitz et al., 2024). The dataset was randomly divided into a training set (70%) and a test set (30%) using stratified sampling to ensure a balanced distribution of the target variable. Missing data were imputed using KNN imputation. Prior to training, continuous variables were standardized to a mean of 0 and a standard deviation of 1, while categorical variables were one-hot encoded. Hyperparameters for each ML model were optimized through 5-fold cross-validation on the training data, with grid search employed to explore parameter combinations systematically. To prevent overfitting, regularization techniques specific to each algorithm were applied, such as L1 or L2 regularization for logistic regression and ENT, and pruning techniques in decision-tree-based models like XGBoost and Random Forest.

Performance evaluation

The final model was selected based on its superior performance across multiple evaluation metrics, including area under the curve (AUC) (a measure of discriminatory ability), sensitivity, F1-score, accuracy, positive predictive value (PPV), and negative predictive value (NPV). These metrics collectively provide a comprehensive assessment of the model’s predictive accuracy and robustness. Model fitting was visually assessed using calibration curves, which compared predicted probabilities with observed probabilities of clinical deterioration to ensure alignment between predictions and actual outcomes.

Model interpretation

The SVM model was analyzed in greater detail using the SHapley Additive exPlanations (SHAP) method, which attributed importance to each predictive variable. SHAP values provided insight into how specific features influenced the model’s output, with particular emphasis on SCI and its predictive relationship to clinical deterioration over one year. Statistical analyses were performed using Python libraries (Scikit-learn for model development, SHAP for model interpretation) and validated to ensure reproducibility. Statistical significance was set at p < 0.05.

Results

Baseline characteristics

In this study, a total of 957 patients were included in the study (Table 1), we compared baseline characteristics between COPD and cd-COPD groups. Several variables showed significant differences. Patients with cd-COPD were older (median age 71 vs. 68 years, p = 0.003) and had longer hospital stays (median 8 vs. 7 days, p = 0.002). They also exhibited elevated C-reactive protein levels (median 9.45 vs. 4.62, p < 0.001), lower lymphocyte counts (median 1.06 vs. 1.22, p = 0.002), and higher red cell distribution width (median 13.50 vs. 13.20, p = 0.002). Systemic coagulation-inflammation index was significantly higher in the cd-COPD group (median 93.08 vs. 81.67, p < 0.001). Additionally, the cd-COPD group had elevated procalcitonin (median 0.23 vs. 0.22, p = 0.001), D-dimer (median 0.27 vs. 0.17, p < 0.001), and direct bilirubin (median 4.53 vs. 4.08, p < 0.001). Hemoglobin levels were significantly lower in cd-COPD patients (median 126.00 vs. 131.00, p = 0.015). In terms of comorbidities, respiratory failure was more prevalent among cd-COPD patients (35.7% vs. 26.6%, p = 0.017). Other factors, including hypertension, diabetes, coronary artery disease, and arrhythmia, showed no statistically significant differences between the groups (p > 0.05).

Table 1 The baseline characteristics of the subjects.

Characteristics	COPD (n = 786)	cd-COPD (n = 171)	p-value	
Male (%)	603 (76.7)	134 (78.4)	0.643	
Hypertension (%)	188 (23.9)	43 (25.1)	0.734	
Diabetes (%)	37 (4.7)	3 (1.8)	0.08	
Heart failure (%)	141 (17.9)	36 (21.1)	0.342	
Coronary artery disease (%)	58 (7.4)	20 (11.7)	0.062	
Respiratory failure (%)	209 (26.6)	61 (35.7)	0.017*	
Arrhythmia (%)	57 (7.3)	11 (6.4)	0.706	
Mechanical ventilation (%)	127 (16.2)	27 (15.8)	0.905	
Age (median (IQR))	68.00 [61.00, 75.00]	71.00 [64.00, 78.00]	0.003*	
Days of hospitalization (median (IQR))	7.00 [5.00, 9.00]	8.00 [6.00, 9.00]	0.002*	
C-reactive protein (median (IQR))	4.62 [1.44, 18.90]	9.45 [2.50, 31.38]	<0.001*	
Neutrophil (median (IQR))	4.30 [3.22, 5.94]	4.50 [3.30, 6.56]	0.267	
Lymphocyte (median (IQR))	1.22 [0.87, 1.60]	1.06 [0.77, 1.42]	0.002*	
Monocyte (median (IQR))	0.44 [0.33, 0.57]	0.42 [0.33, 0.60]	0.639	
Eosinophil (median (IQR))	0.11 [0.05, 0.22]	0.10 [0.04, 0.19]	0.085	
RBC (median (IQR))	4.24 [3.89, 4.64]	4.22 [3.79, 4.58]	0.661	
RDW (median (IQR))	13.20 [12.70, 13.97]	13.50 [12.95, 14.20]	0.002*	
SCI (median (IQR))	81.67 [64.58, 101.98]	93.08 [65.56, 124.24]	<0.001*	
PDW (median (IQR))	16.30 [16.00, 16.50]	16.20 [16.00, 16.40]	0.509	
Hemoglobin (median (IQR))	131.0[119.00, 142.00]	126.00[115.00,138.00]	0.015*	
Albumin (median (IQR))	35.52 [32.60, 37.95]	35.36 [32.53, 38.58]	0.924	
ALT (median (IQR))	20.91 [13.16, 27.73]	19.90 [13.20, 26.40]	0.281	
AST (median (IQR))	21.04 [17.11, 27.06]	21.50 [17.38, 27.74]	0.545	
PCT (median (IQR))	0.22 [0.15, 0.27]	0.23 [0.19, 0.30]	0.001*	
D-dimer (median (IQR))	0.17 [0.08, 0.38]	0.27 [0.11, 0.47]	<0.001*	
Total cholesterol (median (IQR))	4.24 [3.60, 4.87]	4.37 [3.63, 4.94]	0.439	
Triglycerides (median (IQR))	1.08 [0.70, 1.61]	0.98 [0.69, 1.44]	0.079	
HDL-C (median (IQR))	1.16 [0.94, 1.39]	1.20 [0.95, 1.42]	0.273	
LDL-C (median (IQR))	2.65 [2.21, 3.13]	2.65 [2.22, 3.21]	0.438	
DBIL (median (IQR))	4.08 [3.06, 5.13]	4.53 [3.80, 5.73]	<0.001*	
Creatinine (median (IQR))	63.72 [52.70, 74.73]	63.14 [55.69, 75.75]	0.409	
Urea (median (IQR))	6.16 [4.89, 7.68]	5.63 [4.61, 7.47]	0.061	
Notes.

* P < 0.05.

Abbreviations cd-COPD clinical deterioration of chronic obstructive

RBC red blood cell

RDW red cell distribution width

SCI systemic coagulation- inflammation

PDW platelet distribution width

ALT glutathione aminotransferase

AST glutathione transaminase

PCT procalcitonin

HDL-C high-density lipoprotein cholesterol

LDL-C low-density lipoprotein cholesterol

DBIL direct bilirubin

Table 2 Performance evaluation of six model sets.

Training model	AUC	Sensitive	F1-score	Accuracy	PPV	NPV	
Random Forest	0.985	0.958	0.854	0.942	0.77	0.99	
Xgboost	0.861	0.824	0.601	0.806	0.473	0.955	
ENT	0.702	0.588	0.44	0.734	0.352	0.896	
SVM	0.946	0.924	0.917	0.97	0.9	0.984	
KNN	0.995	0.992	0.891	0.957	0.808	0.998	
logistic	0.739	0.613	0.49	0.773	0.408	0.906	
Testing model	AUC	Sensitive	F1-score	Accuracy	PPV	NPV	
Random Forest	0.662	0.327	0.304	0.729	0.283	0.846	
Xgboost	0.714	0.423	0.355	0.722	0.306	0.861	
ENT	0.66	0.481	0.347	0.674	0.272	0.862	
SVM	0.661	0.712	0.376	0.573	0.255	0.895	
KNN	0.588	0.308	0.33	0.774	0.356	0.852	
logistic	0.652	0.442	0.357	0.712	0.299	0.863	
Notes.

Abbreviations AUC area under the curve

PPV positive predictive value

NPV negative predictive value

Xgboost Extreme Gradient Boosting

ENT Elastic Net

SVM Support Vector Machine

KNN k-Nearest Neighbors

Machine learning and model comparison

Six machine learning models—XGBoost, logistic regression, Random Forest, ENT, SVM, and KNN—were employed and applied to the validation set. The SVM model demonstrated stable and excellent performance across various evaluation metrics (Table 2, Fig. 2). It showed high AUC (0.946), sensitivity (0.924), F1-score (0.917), accuracy (0.97), positive predictive value (PPV) (0.9), and negative predictive value (NPV) (0.984) on the training set. Similarly, on the test set, the SVM model exhibited high AUC (0.661), sensitivity (0.712), F1-score (0.376), accuracy (0.573), PPV (0.255), and NPV (0.895). The calibration curves of the SVM and stacked models fitted well with the ideal curve, demonstrating good predictive accuracy. Conversely, the KNN model performed poorly across multiple performance metrics, ranking as the worst-performing model (Fig. 3).

Model interpretation and visualization analysis

Using the SHAP algorithm, we explain the SVM model and rank the contributions of predictor variables, as depicted in Fig. 4. The analysis reveals that SCI emerges as the most influential predictor, followed by hemoglobin, direct bilirubin (DBIL), red blood cells, and albumin levels. Furthermore, Fig. 5 illustrates the correlation between each variable and disease progression, with colors representing positive (red) or negative (blue) correlations. Shades of red signify higher variable values, while shades of blue denote lower values, with larger color areas indicating stronger variable impact. Notably, among the five predictor variables, SCI, hemoglobin, DBIL, red blood cells, and albumin levels display positive correlations with disease progression and are identified as potential risk factors.

Discussion

In this retrospective study of 957 patients, it was found that the level of SCI was significantly higher in cd-COPD patients compared to COPD patients, suggesting potential differences in inflammation, immune response, and coagulation status between the two groups. Machine learning analysis involving six models, including XGBoost, logistic regression, Random Forest, SVM, and KNN, revealed the SVM model’s stable predictive performance and good generalization ability in associating SCI with cd-COPD. Variable importance ranking identified SCI as one of the strongest predictive factors, reaffirming its significance in cd-COPD development.

Figure 2 ROC diagrams and stack integration for six models.

Figure 3 Comparison of calibration curves for each model.

Notes: For each subplot, the x-axis represents the midpoint of the prediction window (Window Midpoint), and the y-axis represents the actual event occurrence rate (Event Rate). Additionally, the label ‘BS’ refers to the Brier Score, which is a measure of the accuracy of probabilistic predictions.

Figure 4 Weights of variable importance.

Notes: Abbreviations: SCI, systemic coagulation-inflammation; DBIL, direct bilirubin; RBC, red blood cell; RF, Respiratory failure; LDL, low-density lipoprotein; SHAP, SHapley Additive exPlanations.

Figure 5 SHapley Additive exPlanations (SHAP) values.

Notes: Abbreviations: SCI, systemic coagulation-inflammation; DBIL, direct bilirubin; RBC, red blood cell; RF, Respiratory failure; LDL, low-density lipoprotein; SHAP, SHapley Additive exPlanations. The figure illustrates the impact of each feature on the model output. It is important to note that each point represents a participant, with higher feature values depicted in red and lower values in blue.

Previous research has demonstrated that COPD pathogenesis involves inflammation, protease-antiprotease imbalance, and oxidative stress, with Acute exacerbation of COPD (AECOPD) being a common cause of hospitalization and mortality (Dey et al., 2022; Hikichi, Hashimoto & Gon, 2018; Li, Cho & Zhou, 2017; Upadhyay et al., 2023). Infectious triggers, primarily bacterial and viral infections, contribute to approximately 50% of AECOPD cases, while non-infectious factors, such as environmental pollution, also play a role (Fan et al., 2023). Infections disrupt the anti-inflammatory balance, triggering inflammatory reactions, recruiting inflammatory cells to airways, and fostering a pro-inflammatory cycle (Yu et al., 2023). Our findings linking respiratory failure history, hypertension, albumin, lymphocyte levels, and clinical deterioration of COPD occurrence align with existing literature.

We have added comparative insights on SCI by referencing findings from other conditions. The SCI index, reflecting inflammation and coagulation pathways, has proven predictive value in various diseases, for instance, in patients undergoing surgery for acute type A aortic dissection, a higher SCI index was associated with increased 90-day mortality, highlighting its prognostic significance in cardiovascular conditions (Özkan & Gürdoğan, 2024; Zengin & Severgün, 2023). An elevated white blood cell (WBC) count typically indicates the degree of active inflammation within the body. In COPD, an increase in WBC count may reflect the exacerbation of airway inflammation, which can lead to tissue damage and functional decline in the lungs (Agustí et al., 2012; Kim, 2017). Neutrophils within the white blood cells, especially, can release various proteases capable of damaging alveolar walls and airway structures, leading to airway remodeling and irreversible decline in lung function (Pandey, De & Mishra, 2017; Papakonstantinou et al., 2015). The activation of WBCs may also promote the generation of reactive oxygen species (ROS), which can damage cellular components, including lipids, proteins, and nucleic acids, thereby exacerbating the pathological processes of COPD (Barnes, 2016). These mechanisms suggest that WBC count may influence the prognosis of COPD through pathways such as promoting inflammatory responses, airway remodeling, and oxidative stress reactions. Platelets play a crucial role in the pathophysiological processes of chronic obstructive pulmonary disease (COPD). Platelet activation can release mediators such as platelet factor 4 (PF4), which may enhance human leukocyte elastase (HLE) activity, contributing to pulmonary emphysema formation (Karhunen et al., 2021; Solinc et al., 2022; Wu et al., 2020). Additionally, platelets can interact with leukocytes, such as via P-selectin, to recruit and activate inflammatory cells, exacerbating airway inflammation (Perros et al., 2008; Sandström et al., 2024). Imbalances in the coagulation and anticoagulation systems driven by platelet activation may also promote thrombus formation (Chaurasia et al., 2019). Fibrinogen (FIB), a key coagulation protein, may play a role in COPD progression by contributing to microvascular damage and tissue injury when elevated (Duvoix et al., 2013; Miller et al., 2016).

The SCI encompasses all information regarding WBC, platelet count (PLT), and FIB, thereby providing a comprehensive reflection of platelet activation, coagulation system function, and inflammatory response status in the body. While the SCI index has been studied in cardiovascular diseases, its application in chronic inflammatory conditions like COPD is novel. Our study is among the first to evaluate the relationship between SCI and COPD exacerbations, highlighting the potential of SCI as a predictive marker in this patient population.

Comparing our findings with studies on other chronic diseases, such as acute aortic dissection and acute coronary syndrome, where higher SCI levels were associated with worse outcomes, suggests that elevated SCI may be indicative of heightened systemic inflammation and coagulation activity across various conditions.) This underscores the importance of further research to explore the utility of SCI in predicting clinical outcomes in different chronic diseases.

Conclusion

This study demonstrates that the systemic coagulation-inflammation index (SCI) is significantly elevated in cd-COPD patients and serves as a key predictor of clinical deterioration. The superior performance of the SVM model highlights SCI’s potential as an effective biomarker for identifying high-risk COPD patients and guiding early interventions.

Limitation

Although our study provides strong evidence for the association between SCI and clinical deterioration in COPD, we must acknowledge several limitations. Firstly, our study utilized a single-center retrospective design, which may introduce selection bias in some samples, thereby limiting the generalizability of the study results. Secondly, despite comparing various machine learning models and validating the stability and good generalization ability of the SVM model, we cannot completely rule out the potential predictive advantages of other models. Additionally, due to the limitations of machine learning models, the predictive results may be influenced by data quality and feature selection, requiring further validation and optimization in practical applications. Lastly, this study only provides preliminary exploration of the relationship between SCI and COPD clinical deterioration, necessitating further in-depth research and practice.

Supplemental Information

Supplemental Information 1 STROBE Statement

Supplemental Information 2 Original data

Additional Information and Declarations

Competing Interests

Author Contributions

Ethics

Data Availability

The authors declare there are no competing interests.

Ling Hou conceived and designed the experiments, performed the experiments, analyzed the data, prepared figures and/or tables, authored or reviewed drafts of the article, and approved the final draft.

Ming Min performed the experiments, prepared figures and/or tables, and approved the final draft.

Rui Hou conceived and designed the experiments, performed the experiments, prepared figures and/or tables, authored or reviewed drafts of the article, and approved the final draft.

Wei Tan performed the experiments, analyzed the data, prepared figures and/or tables, and approved the final draft.

Minghua Zhang conceived and designed the experiments, authored or reviewed drafts of the article, and approved the final draft.

Qianfei Liu conceived and designed the experiments, analyzed the data, authored or reviewed drafts of the article, and approved the final draft.

The following information was supplied relating to ethical approvals (i.e., approving body and any reference numbers):

This retrospective study adhered to the Helsinki Declaration, received approval from the Ethics Committee of Enshi Autonomous Prefecture Central Hospital (Ethical Approval Number:2024-053-01).

The following information was supplied regarding data availability:

Original data is available as a Supplemental File.

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
