# Peer review of "Prediction of clinical deterioration within one year in chronic obstructive pulmonary disease using the systemic coagulation-inflammation index: a retrospective study employing multiple machine learning method"

_PeerJ, doi:10.7717/peerj.18989_

## Round 0.1 · original submission · Major Revisions

Please address all the comments from both reviewers

Reviewer 1 ·

Basic reporting

The title effectively lists the outcomes measured and patient population as well as the design. The Abstract is well organized and the text is sound, yet I recommend that you
a) list the mean age and sex breakdown of the patient cohort and
b) the word ‘notable’ should be replaced with significant. If you are denoting p values, which you do, the proper word needs to be used here.
I assume that ‘cd’ refers to clinical deterioration yet this is not denoted here, so please do so.
The test stated that these models are ‘evaluated,’ but how? Please briefly denote this here.
Please consider reporting data for the SCI here, especially if this is the main finding of this manuscript.

Introduction
• Ln 44: The notation of cdCOPD is good, yet this needs to be spelled out here, then abbreviated.
• Ln 52-54: This is good text, yet it would be strengthened by adding some detail (data from various studies) to support this point. E.g. For example, data from () in ____ (patient population) suggest that…
• Ln 55: Please add a reference here regarding the SCI and some prior data showing its range of values to provide the reviewer-reader with more information regarding its use and efficacy. If you strengthen this section, then this text concerning what you hypothesize will follow better from the Introductory text presented here.
I recommend that you move the hypothesis to the last line of this section, and then after the hypothesis, add a brief sentence denoting what the broad application of this study is to Medicine.
Overall, the Introduction is sound, yet it would be improved by adding some new text to better explain various terms and strengthen the study rationale. The manuscript will benefit from copy editing to enhance the clarity of the manuscript.

Figures & Tables
The tables are very clear and well structured, as are the Figures, yet please make sure to add proper abbreviations and make the Figures bigger so they are more legible.

Experimental design

Material and Methods
• Ln 68: Please clarify what the term ‘severe’ means here; it may benefit the text to be more clear and objectify this term.
• Ln 71-72: Please provide a rationale for this 70-30 split of training to testing of the samples(data).
• Please consider denoting why only 3.5 yr of patient data were acquired in this study.
• Ln 85-87: It would be more clear to state which specific outcomes were compared with the t test, etc. and this should be an independent t test due to the 2 unique groups you are comparing with this statistic.
• Ln 89-91: Please denote why these 6 models were used—if these are typical models used in the field or in the(your) lab, validate them with a proper reference citation.
• Ln 95: It is not clear which criteria were used to identify the ‘best’ model, so please be more clear here with this text.
• Ln 97-101: Similar to my comment above, it would benefit the text if a reference citation is added here to validate this approach. It is not guaranteed that a Reviewer will be more familiar with this method, and attributing it to a source will somewhat validate its suitability here.
I do not have any concerns related to the research, yet this section would benefit from greater clarity and explanation of these methods which can be done with pointed addition of next text in this section.

Validity of the findings

Results
• Ln 110: This result is not statistically significant, so please soften your language here.
• Ln 117: Please add (p > 0.05) here to reflect that none of these variables is significantly different between groups.
• Ln 133-141: if applicable, it would benefit this text to denote the strength of these correlations here. Then in the Discussion, please evaluate the magnitude of these associations based on prior literature.
• Ln 142-144: I recommend that this text be deleted as this is explaining the findings, which best fits in the Discussion text.
• The Results text is very clear and this section is well organized.
• In table 2, please spell out all abbreviations used to include SVM, etc.
• In Fig 3, please label the x axis and spell out what BS represents, to be more clear to the reader.
Overall, these results are novel and will build on knowledge in this area of study.

Discussion
• Ln 174-182: This is an interesting section of text, but please re-organize this in sentence form rather than in a numbered list, which is awkward. Also, you are not measuring platelet counts in this study, so this text may seem like excess speculation or may be viewed as straying from the study and its actual outcomes. Please consider condensing this section of text.
• Ln 183-192: Similar to my comments above, this is well structured text but some language needs to be added to make it clear to the reader that this section concerning fibrinogen is relevant to your study. This reads more like a literature review than (proper Discussion) text which has the aim to explain and interpret your results vs. prior literature.
• Please consider comparing your SCI values to those of other groups of patients with chronic disease. In fact, re-reading your Discussion concerning SCI shows no text where your data are evaluated vs the literature—this is a weakness of this section and should be addressed.
• Ln 196:197: Please delete this statement as this belongs in your Introduction and not here where you should be evaluating your SCI data.
Also, no text evaluates your data from Figs 4 or 5; please consider adding Discussion text to address this.
Overall, the Discussion’s focus strays from the need to interpret their data vs existing literature and strays too much towards speculation and has excess text on outcomes which were not measured, such as platelet count and fibrinogen. The fact that these were not assessed is a study limitation which is not addressed by the authors.

Conclusion
No Conclusion is provided with the submission, which is a weakness. Please add a brief Conclusion section restating the primary findings and how they apply to Medicine.

Additional comments

no comment

Reviewer 2 ·

Basic reporting

1. Please define abbreviations the first time they appear in the text. For example, cd-COPD was not defined. COPD was not defined the first time it appeared. SCI was not defined the first time the word appeared. Authors should carefully make correction to abbreviations so as to help readers navigate the manuscript with ease.
2. A clear hypothesis needs to be in the introduction.

Experimental design

1. The manuscript needs more information on how ML algorithms were implemented. Currently, there is not sufficient information provided to replicate the analysis by the reader or the rest of the scientific community.

Validity of the findings

1. SCI levels have been previously linked with COPD severity and risk (e.g., https://doi.org/10.1371/journal.pone.0303286). These are very closely connected with COPD exacerbation outcomes and clinical deterioration in COPD (which is persistent AECOPD?) I think lines 196-197 need to be modified to reflect this fact.
2. Expanding on the point 2, the authors elaborate on the importance of platelets and leukocytes. Numerous association studies link other leukocytes to COPD exacerbation susceptibility (e.g., https://link.springer.com/article/10.1186/s12931-020-01436-7). The author should discuss where the SCI index fits with other observational studies.
3. Is cd-COPD just another term for severe exacerbation of COPD?
4. Did the admission code only define the acute exacerbation event? Has any attempt been made to determine medication use (antibiotics and/or oral steroids) or apply the Rome Proposal Definition of COPD Exacerbation (PMID: 34570991)?

Additional comments

1. Line 88: Please define which data were skewed.
2. Gender should be defined as biological sex unless a specific question was asked to the patient about their gender identity.

---

## Round 0.2 · accepted · Accept

The authors have addressed all of the reviewers' comments and the revised manuscript is now ready for the publication.

Reviewer 1 ·

Basic reporting

Title & Abstract
The title is adequate as written; as for the Abstract, there are still a few abbreviations (RDW, etc.) which need to be spelled out for clarity. Also, you use the term ‘significant disparities’ here yet no p values are denoted here, which is necessary. So, please add p < 0.05 or p < 0.01 here to reflect this result.

Introduction
This revised section of text is much improved than the initial iteration, as the study background is strengthened, allowing the reader to better understand this topic and why this study is needed.
The language is adequate, although very technical and full of a lot of jargon, which
makes it somewhat hard to read and understand.

Figures & Tables
Per my original comment, abbreviations have been added to the tables and the figures are now larger in size, which overall enhances their clarity.

Experimental design

Material and Methods
The revised Method is much more clear in describing the participants and protocols used in this study; however, the Authors still have not identified which specific variables are analyzed with the t-test, and in addition, whether a paired or unpaired t-test was used in the data analysis. This is important and fundamental information which needs to be denoted here, and in any Statistical analysis section, to be completely transparent to the reader. As I had previously stated, if you are comparing various demographic traits between the cd and non-cd COPD groups, you would use an unpaired t test as these scores do not depend on each other.

Validity of the findings

Results
Please make sure that all data presented in text (page 5, paragraph 1) have proper units, such as for hs CRP, blood glucose, etc.
The SVC model for the test set has a PPV = 25.5%--this value is relatively low, correct? If this is the case, please revise this text to reflect this, as I believe that 100 % is a ‘high’ (maximal) value.
Other than these items, the Results is well-written. However, cannot judge the plausibility of these results as presented here.

Discussion
This revised section of text is vastly improved, especially content related to platelets and fibrinogen which now shows a link to COPD risk or onset. There is much greater comparison of current results to prior data, which is appropriate in this section of text. Lastly, there is text which suggests how these results apply to the broader field of medicine and treatment of COPD, which is noteworthy.

Conclusion
The Authors have included a Conclusion with this iteration, and it effectively denotes the study take home message.

Additional comments

NA

Reviewer 2 ·

Basic reporting

I am pleased with how the authors responded to all my comments. They have effectively addressed each of my questions. I have no additional comments to offer.

Experimental design

I don't have any more comments to add.

Validity of the findings

I don't have any more comments to add.

Additional comments

I don't have any more comments to add.